# Octopus track chemosensory plumes to find food

Willem Lee Weertman[1,2,3*], Venkatesh Gopal[4], Dominic M. Sivitilli[1,2‡], David Scheel[3], David H. Gire[2]

1 University of Washington Friday Harbor Laboratories, Friday Harbor, Washington, United States of America, 2 Department of Psychology, University of Washington, Seattle, Washington, United States of America, 3 Institute of Culture and Environment, Alaska Pacific University, Anchorage, Alaska, United States of America, 4 Department of Physics, Elmhurst University, Elmhurst, Illinois, United States of America

‡ Current Address: University of Washington Applied Physics Laboratory, Seattle, Washington, United States of America

* willemlw@uw.edu

**Data availability statement:** The data underlying the results presented in the study are available from (https://github.com/weertman/OctopusUseOdorPlumes).

## Abstract

Chemosensory plume-guided navigation, tracking a chemical plume to its source, is a primordial behavior used by many animals to search beyond the visual range. Here we report the first laboratory observations of octopuses performing this behavior, demonstrating that they can use chemosensory plumes to find food. In a three-station discrimination task carried out in the dark, octopus showed a strong preference to move upstream towards the food-baited target, supporting the hypothesis that they are performing chemosensory plume-guided search. When seeking a single baited target, also in the dark, octopuses not only preferred to move upstream towards the food source, but they also displayed characteristic motions associated with odor-gated rheotaxis, a commonly used chemosensory tracking strategy used by many animals, which includes pausing, switchbacks, and across-stream redirections to the bait. Additionally, when approaching single baited stations the octopus often made fast reactive lunging motions. The observation of these fast *arm-aligned* motions (FAAM), taken together with the observation that the octopus did not have a characteristic body axis orientation to the bait, as would be expected if bilaterally symmetric organs such as the olfactory organs guided this behavior, supports the hypothesis that the suckers are the primary chemosensory organs driving octopus chemosense-guided behaviors.

## Introduction

Efficient use of chemical cues for search and way-finding is an evolutionary pressure experienced by most organisms [1,2]. Despite having a sophisticated, highly divergent chemosensory system, and complex behaviors, cephalopod chemosensation has received limited, but growing, modern scientific attention. Coleoid cephalopods are often left out of cross-phyla comparisons in favor of simpler Mollusks [2–4]. Currently, there is debate over the extent to which octopus use distant chemical cues to find food at all [5]. Multiple detailed observations of octopus chemosensory behaviors have been made [5–14], yet, to date, there have been no

**Funding:** WLW was significantly aided by the support of the Beatrice Crosby Booth Endowed Scholarship at Friday Harbor Laboratories, Alan J. Kohn Endowed Fellowship, and crowdfunding on Experiment.com. VG would like to gratefully acknowledge generous support from the Hyslop-Shannon Foundation. DSc acknowledges support from an Indigenous One Health development grant from an Alaska Native & Native Hawaiian Serving Institutions Program of the U.S. Department Of Education.

**Competing interests:** The authors have declared that no competing interests exist.

such detailed descriptions of octopus chemosensory plume-guided search behaviors. Among cephalopods, this behavior has only been studied in nautilus [15,16].

In a recent review of chemosensation, Mollo *et al.* [17] put forward a compelling proposal to develop a unifying language with which to describe chemosensory behavior, and we have implemented their suggestions here. Before we proceed further we would like to clarify some terminology that we use in the remainder of the paper. Chemosensation, or chemical sensing in organisms is the interaction of 'odorant' molecules with sensory receptors that leads to behavioral outputs. Chemosensation occurs in aerial, terrestrial, and aquatic environments, and involves molecules that can be hydrophobic or hydrophilic, and volatile or non-volatile. Thus chemosensory behavior can, and does, occur in an enormous range of combinations of ecological environments - terrestrial, aquatic, aerial - and organisms, that can range from single-celled to multicellular. As pointed out by Mollo *et al.*, this leads to "a vast and blurred variety of modes of chemical communication that could be collectively called "chemosensation," which always starts from the interactions between ligands and receptors, two chemical entities both occurring in an immense structural variety in nature [17]." However, perhaps biased by studies of humans, the literature has been split by the "touch-taste dichotomy" [17] in which olfaction is considered a 'distance sense' like vision or audition, and taste is thought of as a 'contact sense' like touch, which has created "an "epistemological obstacle" pervading the chemosensory literature: the ancient belief that taste and smell are two different senses." Mollo *et al.* argue strongly, and in our opinion, convincingly, that the touch-taste dichotomy does a disservice to the field of chemosensory studies as it imposes an artificial, and incorrect, binary split on chemosensation studies, and that this "unnatural categorization of chemosensory processes ... prevents the development of a satisfactory narrative on the evolution of chemical communication." Further, these contradictions and ambiguities are particularly prevalent in aquatic environments [17–20], which is the environment we have considered in this study. In particular, in recent years there has been a great deal of interest in studying how organisms perform long-range 'odor following,' 'odor tracking,' or 'odor navigation.' However, motivated by [17], we instead use the terms 'chemosensing' or 'chemosensation' instead, except in cases where we are referring to the literature, where it would be confusing not to use the terms that have been used in the original papers being cited or discussed. In particular, *odor-gated rheotaxis* is a term that is widely used in the literature, and we use the same term in our paper instead of referring to it as *chemosensation-gated rheotaxis*.

Chemosensory-guided navigation in turbulent flow environments is one of the most remarkable examples of long-range search carried out by animals. Understanding this behavior is a challenging problem in sensory biology (Fig 1d) because turbulence rapidly breaks up the chemical plume into sparse and intermittent chemical patches that the animal encounters very infrequently [21]. In addition to detecting a signal that is intermittent in time, the task is further complicated because the animal is also under 'time pressure' as it must often navigate to the chemosensory source quickly to reach the resource before a competitor does. While it may seem that waiting to build up a time-average of the chemosensory signal might be an effective search strategy, it has been shown that time-averaging does not converge quickly enough to be useful [22].

Two important results have emerged in recent years that have had a great bearing on the scientific understanding of olfactory search. The first is that the spatiotemporal structure of the chemosensory plume is determined by the physical environment in which the flow takes place, which has led to the realization that "the physical environment dictates the navigational behavior employed by an organism" ([23], see also [24,25]). This is especially true in the benthic zone that the octopus inhabits. Here, surface friction due to the often rough substrate causes shear, which in turn amplifies the turbulence close to the substrate [23,26,27].

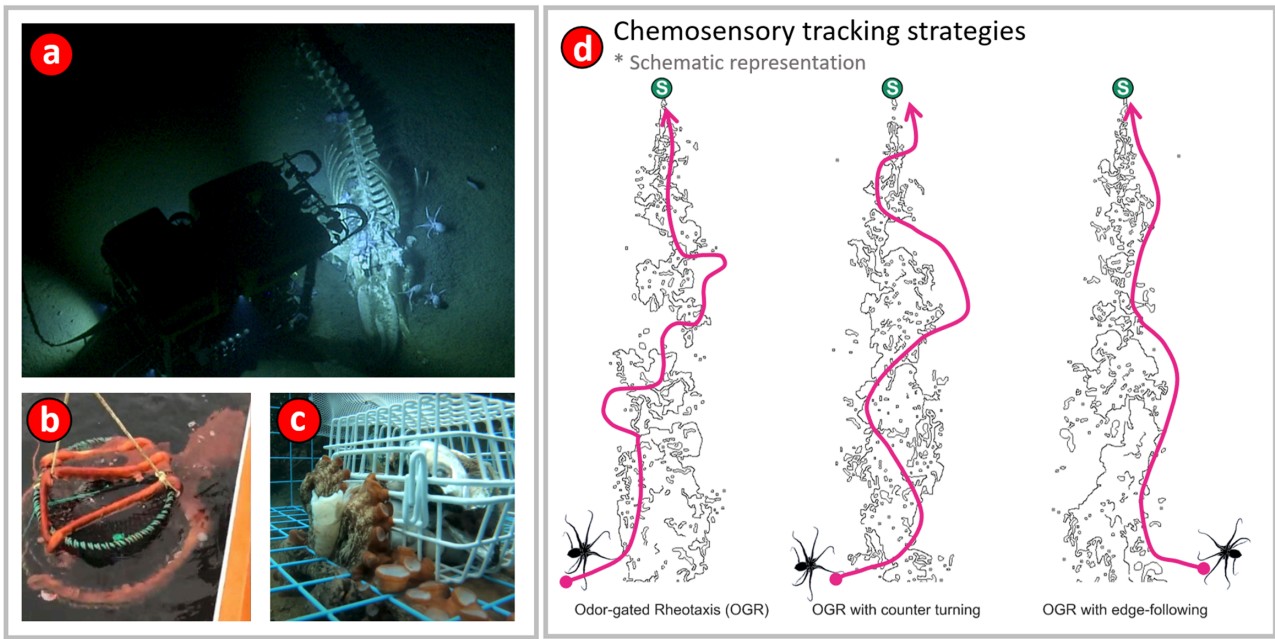

**Fig 1. (a) Indirect evidence for long-range chemosensory tracking by octopus.** Frame from a video recorded at a depth of 3,238 m near the Davidson Seamount in the Monterey Bay National Marine Sanctuary by the Ocean Exploration Trust vessel EV Nautilus (United States National Ocean and Atmospheric Administration (NOAA), 2019). Numerous *Muusoctopus robustus* feeding on the carcass of a whale are visible, lower right. At this depth, which is far below the photic zone of the ocean, it is unlikely that octopus can use vision-guided search to arrive at the whale fall, thus chemosensory-guided navigation is the likely search strategy employed. **(b)** Giant Pacific Octopus (GPO) attached to a spot prawn trap off the coast of Vancouver Island, British Columbia, Canada. Spot prawn traps are baited and placed on the ocean floor, typically at a depth of 130 - 160 m. **(c)** An octopus eating bait from a baited fish trap. **(d) Strategies for Odor Gated Rheotaxis** The basic mechanism in odor gated rheotaxis (OGR) is that the animal 'surges' upstream (or upwind) on encountering a patch of odorant, and then 'casts,' or moves in a zig-zag manner perpendicular to the direction of the mean flow to regain contact with the chemical plume. Several variations on this basic strategy are possible, two of which are shown in the figure. In OGR with counter-turning the animal moves upstream and diagonally across the width of the plume and counter turns into the plume on encountering the plume edge. In OGR with edge-following the animal stays on one side of the plume centerline and makes counter turns to locate a single edge of the plume while also moving upstream. We observed possible evidence of all three mechanisms by octopus during chemosensory tracking. It should be noted that the fine black lines represent a simulated theoretical odor plume and the magenta line is a schematic representation of the animals' hypothetical trajectory under the three different strategies.

The second is that the rapid temporal fluctuations of the chemical concentration carry directional information that the animal can use to orient towards the source [28–31]. Thus, the faster the neural response of an animal sensory system, the more it can take advantage of the spatiotemporal information contained in the concentration fluctuations of the chemosensory plume.

Octopuses are benthic cephalopods occupying habitats that span a wide variety of flow regimes, from shallow tidal pools, to the ocean floor at a depth of over 3,000 meters. It is in this context – existing in a turbulent flow environment and possessing a fast neural system capable of signal transduction on millisecond timescales [32] – that the chemosense-guided search behavior of the octopus should be considered.

Although chemotactile, or 'touch by taste' sensing [33] is clearly an important specialization used by foraging octopuses, their use of distance chemoreception to locate or identify food has received little investigation. Octopuses actively forage in spatially complex environments (e.g. crevices in reefs), often for live, mobile (e.g. Crustacea, Gastropoda) or sessile (e.g. some Bivalvia) prey. Octopuses are typically not scavengers except opportunistically. While visual stimuli can be sufficient to elicit predatory attack, visible movement by a target is not

necessary to invoke this response [34]. Despite the role of vision in some predation, octopuses can detect food chemicals [14], and rely more on chemotactile information than on vision in food choice [35]. On broader spatial scales of the order of several body-lengths, only octopus visual and chemotactile foraging has been studied, and the important role of distance chemoreception in large-scale navigation has not been systematically investigated. However, there is a great deal of anecdotal indirect observations of long-range chemosensory plume-guided search by octopus, which includes (but is not limited to) a sighting of many *Muusoctopus robustus* feeding on a whale fall by RV Nautilus in 2019 (Fig 1a), and the common occurrence of octopus in deep-set baited traps (Fig 1b,c) observed by fishermen worldwide [36].

Some mollusks use chemoreception in predator avoidance, although this has yet to be demonstrated for octopuses. For example, both scallops [37] and gastropods (See [38] and references therein) increase escape responses in the presence of chemicals associated with nearby predator activity. However, both octopuses themselves and their predators move considerably faster than scallops, snails, and some of their common predators (e.g. sea stars), and a reliance on chemosensitivity in predator avoidance may not be as reliable as visual cues. The use of chemosensation in predator avoidance has been shown in bobtail squid [39], and it would not be surprising if octopus also used chemosensation in addition to vision for predator avoidance. Chemoreception and plume tracking likely do play a role in the reproductive biology of octopuses. Walderon *et al.* [14] established that *Octopus bimaculoides* are sensitive to chemical cues from distant conspecifics, and that female *Hapalochlaena maculosa* are sensitive to chemical cues emitted by males, possibly using this information to influence mate choice [40]. Some male octopuses possess enlarged suckers on the forward arms, possibly used in display [41], but which are also equipped with chemoreceptive cells around the sucker rims. Cosgrove and McDaniel [42] observed male *Enteroctopus dofleini* positioned atop rocks, and spreading their arms into the current while slowly turning back and forth, behaviors presumed to facilitate chemosensory detection and plume tracking. Further, both Cosgrove, and McDaniel and Anderson [43] observed multiple male *E. dofleini* surrounding mature females, presumably attracted by chemical signals.

We describe two laboratory experiments which found evidence that *Octopus rubescens* does use chemosensory plumes to find food. *O. rubescens* is a small cold-water species of octopus local to the Salish Sea. The experiments, a discrimination task and an approach task, were run using a first of its kind, octopus-safe, open-circuit seawater flume, in the dark under near-infrared lamps. The discrimination task was carried out to prove that octopus used chemosensory plumes to find food, and the approach task was used to study how the octopus moved during chemosensory tracking.

## Materials and methods

### Ethical animal care

*Octopus rubescens*, which was used for this study, is a cold-water species of octopus local to the Salish Sea. All octopus were collected from the same location in Admiralty Bay, Washington State, by American Academy of Underwater Sciences authorized divers under a permit approved by the Washington Department of Fish and Wildlife. These octopuses were transferred to, and housed and cared for in the open circuit seawater system at the University of Washington Friday Harbor Laboratories. Animal care was done in accordance with a protocol approved by the University of Washington Institutional Animal Care and Use Committee (Protocol number: 4356-02). Octopus diet during housing and experiments consisted of both

live and fresh, raw, locally collected species of small shrimp and crab (*Pandalus sp.* and *Hemigrapus sp.*). Prior to behavioral experiments, to ensure food preference, for at least a week the octopus were fed the same species of fresh raw shrimp or crab species that they would be fed as a reward during experiments.

## Flume design and water flow characteristics

The flume was constructed from acrylic and rested on a stand made from black t-slotted aluminum (Fig 2c). Beneath it, a matte black, treated plywood sheet served to optimize image contrast (Fig 2a,b). The experimental area within the flume measured 185 cm in length (in the direction of flow) and 116 cm in width, with a water depth varying between 10 and 12 cm across its length. The height-to-width proportion closely follows the optimal height-to-width ratios recommended by Nowell [26] to effectively simulate naturalistic benthic turbulent flow conditions.

To equalize the pressure distribution inequalities caused by the intake pumps and standpipes, and to contain the octopus, a series of baffles were placed at both the upstream and downstream ends of the flume's experimental space (Fig 2c). The innermost baffles were designed with 2.54 mm holes spaced 12.7 mm apart to prevent octopuses from squeezing through them. Additional precautions to prevent escape included an extra 30 cm of wall height above the water surface and a double inset rim with a downward-facing mesh lip. To exclude all external light for experiments, a blackout tent encased the flume. The tent was constructed with a PVC frame and lined with 7 mil Panda film black-side turned in. During experiments, illumination was provided by eighteen 850 nm near-infrared illuminators positioned around the flume's perimeter and three 950 nm near-infrared LED strips beneath the flume to enhance backlighting. Additionally, four smart LED lights (Philips Hue White) were placed at the four corners of the flume to minimize surface glare, and were controlled via Bluetooth (Philips Hue Hub) to simulate natural light cycles from 0530 to 1730 hours with slowly dimmed, or brightened, one-hour-long artificial sunrises and sunsets.

Water flow within the flume was driven by two variable fixed-speed controlled pumps (EcoTech Marine, Vectra L2), submerged in an open-circuit seawater system reservoir and plumbed to the flume. Set to their minimum setting, these pumps circulated an estimated 200 liters per minute of fresh seawater through the flume during the experiments. Flow characteristics within the flume were maintained with periodic cleaning of the baffle plates, fixing the pump settings, and keeping a homogenous interior.

## Octopus pose estimation using DeepLabCut

The octopuses were recorded continuously at a rate of 10 frames per second with a 1920 × 1200 monochromatic USB 3.0 camera (Basler a2A 1920 160umBAS), which had its near-infrared filter removed and was controlled through a custom Python script using the open source PyPylon library. To track the location of octopus within the flume DeepLabCut markerless pose estimation models were used [44]. A model was trained which detected the location of the eyes and mantle tip of the octopus in the flume (Fig 2a,b). Octopus eye coordinate predictions can be used to estimate the body axis orientation and the center of the octopus (Fig 2a,b). Three versions of the model were trained to 500k iterations using the machine label correction tool provided by DeepLabCut to update the dataset between versions. The final model reached a test mean squared error eye location prediction accuracy of 6 mm within the 185 x 116 cm flume, a mean error smaller than the size of the octopus eye. Following eye coordinate tracking, to clean the data, statistical outlier coordinates were detected, dropped,

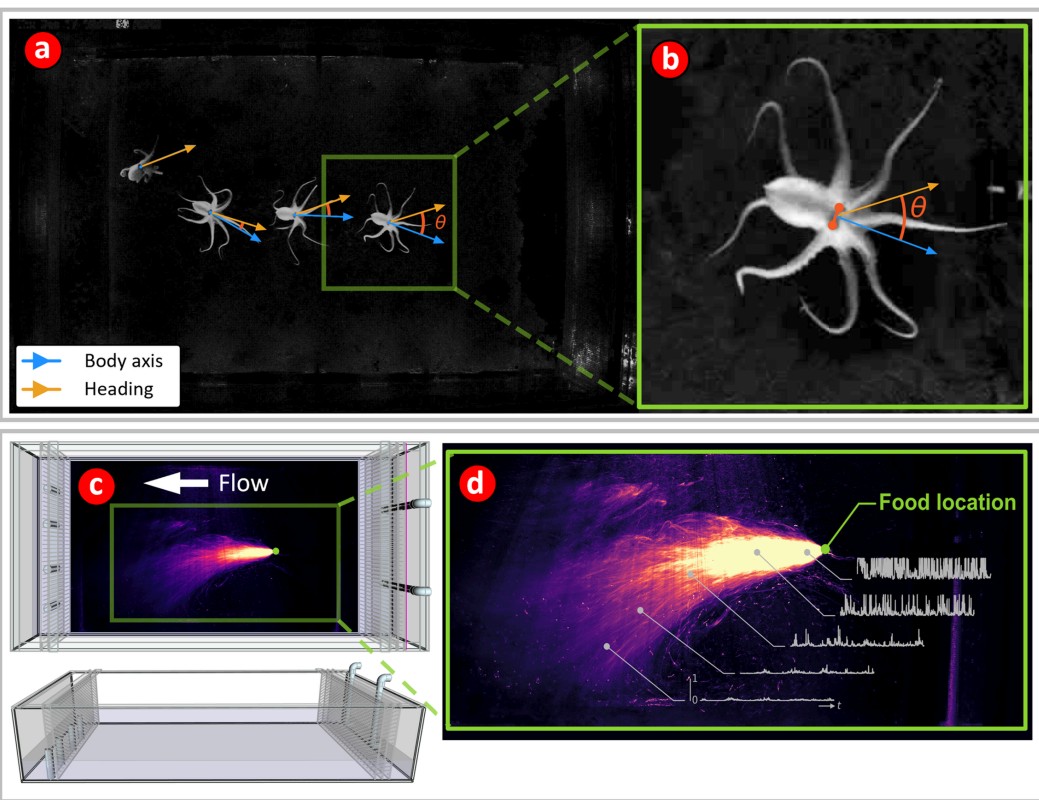

**Fig 2. (a, b) Octopus motion tracking - Principal vectors.** Octopus motion was primarily quantified by two vectors - the body axis vector (blue), and the heading vector (light orange). (b) The angle $\theta$ between the body axis and heading vectors is the heading offset angle. A DeepLabCut two-dimensional pose estimation model was trained to locate the eyes of the octopus in the video data (shown in (b) as solid orange dots). The midpoint of the line connecting the two eyes was denoted as the 'body center' and was the reference point used to compute kinematic quantities. The body axis vector (blue), is a unit vector that has its base located at the body center, and points in a direction away from the mantle and perpendicular to the line joining the eyes. The heading vector is a unit vector that points in the direction of motion of the body center between two successive video frames. Thus, the heading vector also points in the direction of the velocity of the body center. **(c) Flume schematic diagram showing top and side views of the water flume used for the behavioral experiments** The flume was 185 cm long and 116 cm wide with a water height between 10 - 12 cm. The water flow was from right to left at a speed of approximately 2 cm/s. Superimposed on the top view is a time-averaged visualization of a simulated chemosensory plume relative to the flume area. **(d) Visualization of time-averaged simulated chemosensory plume** The heat map shows the time-averaged intensity of light scattered from a simulated chemosensory plume, showing the approximate extent of the plume. The chemosensory plume was simulated by seeding the flow with 9-13 micron diameter glass beads. A syringe pump injected a dilute suspension of the beads approximately iso-kinetically into the flow at the location of the food source. The flow is slightly biased to the bottom of the page because, due to floor drainage and surface ruggedness, it was not possible to level the tank exactly. Typical time traces of the scattered intensity are shown at different locations in the plume demonstrating how chemical concentration fluctuations are likely to vary at increasing distance from the chemosensory source.

and then linearly interpolated between. After interpolation, a 5th order Savitsky-Golay (Savgol) filter with a window size of 29 frames was used to smooth the data. Cleanliness of the tracking jitter was considerably improved by the Savgol smoothing (App. 1, Fig 1; App. 1, Fig 5). Useful extractable features from octopus eye coordinate locations are the average eye position, which is located at the midpoint of a line joining the centers of the two eyes, the body axis vector, which is perpendicular to the line drawn between the eye centers and facing away from the mantle tip, the average eye-position velocity vector, and the angle between

this velocity vector and the body axis vector, which is hereafter called the heading offset angle. This angle is a measure of the angular offset between the direction the octopus is facing and the direction in which it is moving. The octopuses in the study had different sizes, and consequently different speed distributions. Speeds were normalized by using z-scored speeds (subtracting off the mean speed and scaling by the standard deviation) instead of raw speed values (App. 1, Fig 2).

## Octopus arm location estimation using a custom image processing method

Octopus arms are difficult for pose estimation models to track because of how small the arm tips are and how quickly they move. An attempt to track all eight arm tips was made, and after 11 iterations of DeepLabCut model training on 2,200 labeled images, the test mean squared error of the arm tips plateaued at around 18 mm, a size much greater than that of the arm tips which are approximately 1 mm across. This error was primarily observed to be a failure of the model to tell which arm tip was which, an error which is difficult to correct due to how the arms move relative to each other (App. 1, Fig 1). Because of this limitation, we developed a geometry-based computer vision method which used the high contrast images of the octopus in the flume and the DeepLabCut eye location predictions to detect arm positions as peaks in pixel intensity within a circle around the octopus relative to their body axis. This algorithm was only used to assess arm motions during the single station approach task described below.

The first step towards estimating arm location is a dynamic background subtraction procedure in which the background is subtracted from each frame by locating the ten nearest frames in time in the video in which the octopus is more than 600 pixels away from its current position. These ten frames are averaged and used as the background for the current frame. This dynamic background subtraction is required because the open circuit system rapidly deposits sediment on the flume floor, which causes the background to change frequently. In the next step, $600 \times 600$ pixel, centered and cropped images of the octopus are extracted from all background subtracted video frames. These images are then rotated to a standard position in which the body axis of the octopus points vertically downwards in all the images. Then, for each individual octopus, all the images have their pixel-wise average pixel intensities computed creating an average body-centered image of the octopus (Fig 3c). The average image is then thresholded, masked, and a circle is fit to the corners of the mask bounding box which are furthest from the octopus' eyes (Fig 3a-c). This circle provides a ring around the octopus from which pixel intensities can be grabbed in a size-normalized manner that will include the arms but never the mantle (Fig 3d). Given the varying contrast of images of the octopus in this experiment, this method captures the motion of the octopus arms with minimal background noise.

This ring (the thick gray circle in Fig 3d) is then divided into five circles, each of which is one pixel in width. These circles are unwrapped into 1D arrays and then binned into 100 angular bins of average pixel intensity (Fig 3e) resulting in a 5 x 100 array. After this, bin-wise averages are taken resulting in a 'flattened' $1 \times 100$ array representing the mean pixel intensity values of the circle for each image. We used the SciPy `find_peaks` algorithm to find the peaks corresponding to the location of the arms. Due to the circular nature of the data, peaks lying on an edge of the array are not detected by the algorithm as the peak gets split into two parts, which end up on opposite ends of the array. For example, a peak that lies a few degrees above and below zero will have one part of the peak at zero and the other at $2\pi$. To ensure that such peaks were properly detected, the circular data array was shifted left or right by a short amount and the peak finding algorithm was run on the shifted data. Detected peaks were then assigned into an array of 100 angular bins, each 3.6° in width. The angles between the detected

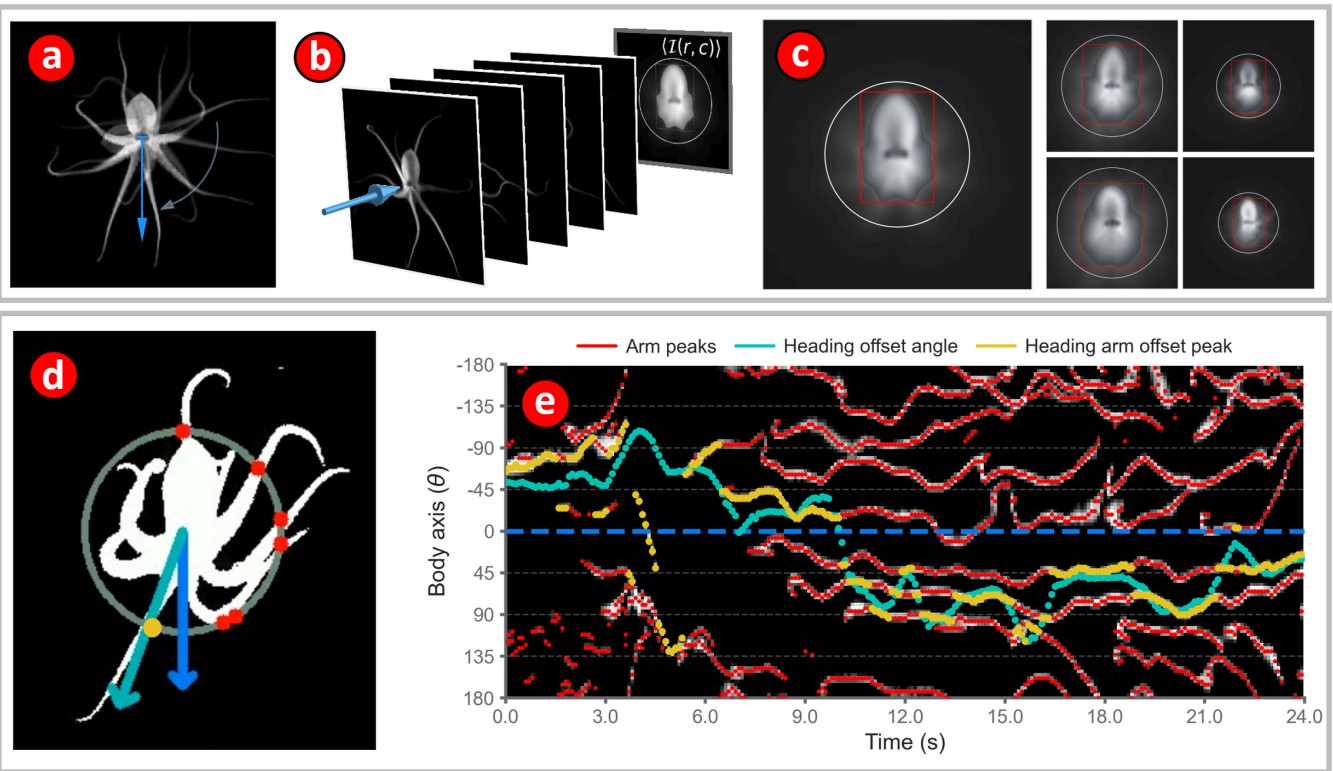

**Fig 3. (a, b) Estimating direction of octopus motion relative to arm position.** DeepLabCut predictions of eye coordinates allow for the creation of a body axis vector (blue) that can be used to quantify the orientation of the octopus. The body axis vector has its base at the center of the line joining the two eyes and points perpendicular to this line in a direction pointing away from the mantle. For each octopus, all images were first rotated to a standard orientation with the body axis pointing vertically downwards as shown in (a). A normalized body image was then generated by averaging all images as shown in (b), where the blue arrow shows the stack of images that are added and scaled by the maximum summed pixel intensity to produce the normalized body image (where, $r$ and $c$ represent row and column, respectively, and $\langle \mathcal{I}(r, c) \rangle$ denotes the normalized intensity). **(c)** Normalized body images are then used to produce a unique bounding rectangle and bounding circle for each octopus. First, a rectangle that is symmetric about the body axis of the octopus is generated. The extent of the rectangle is such that it contains 95% of all the intensity in the image. Then a circle is drawn with its center at the midpoint between the eyes and its radius equal to the distance to the most distant vertices of the bounding rectangle. Variations in octopus size can now be adjusted for by normalizing the radii of the bounding circles to unity. Bounding rectangles and circles for five different octopuses are shown. **(d)** Arm positions were estimated by determining the points of intersection of the arms with the bounding circle for that octopus using the NumPy `find_peaks` algorithm on background subtracted images of the octopus (red dots). The arm that is most closely aligned with the direction of motion, or heading, is termed the heading arm. The angle between the heading arm and the heading vector is called the heading-arm offset angle. **(e)** Quantifying arm position allows us to track the heading offset angle.

peak bins and the octopus heading were then computed; these angles are hereafter called the *heading arm offset* as they are the angle between the direction of the octopus's motion and the arm locations. Following calculation of heading arm offsets for all peaks, the smallest heading arm offset for each frame, that is, the arm closest to the direction of motion, was kept to assess fast arm-aligned motions (FAAM) of the octopus (Fig 3e).

## Three-station discrimination task

*O. rubescens* (3 male, 2 female) were each tested nightly for a week on a three-station discrimination task (Fig 4). Every night before the artificial sunset, which started at 17:30, the octopus were transferred into the flume while within their dens. The octopus was not exposed to the flume before the experiment. Their dens were constructed of infrared-transmitting acrylic so the cameras could see the octopus inside the den. An hour after artificial sunset at 18:30

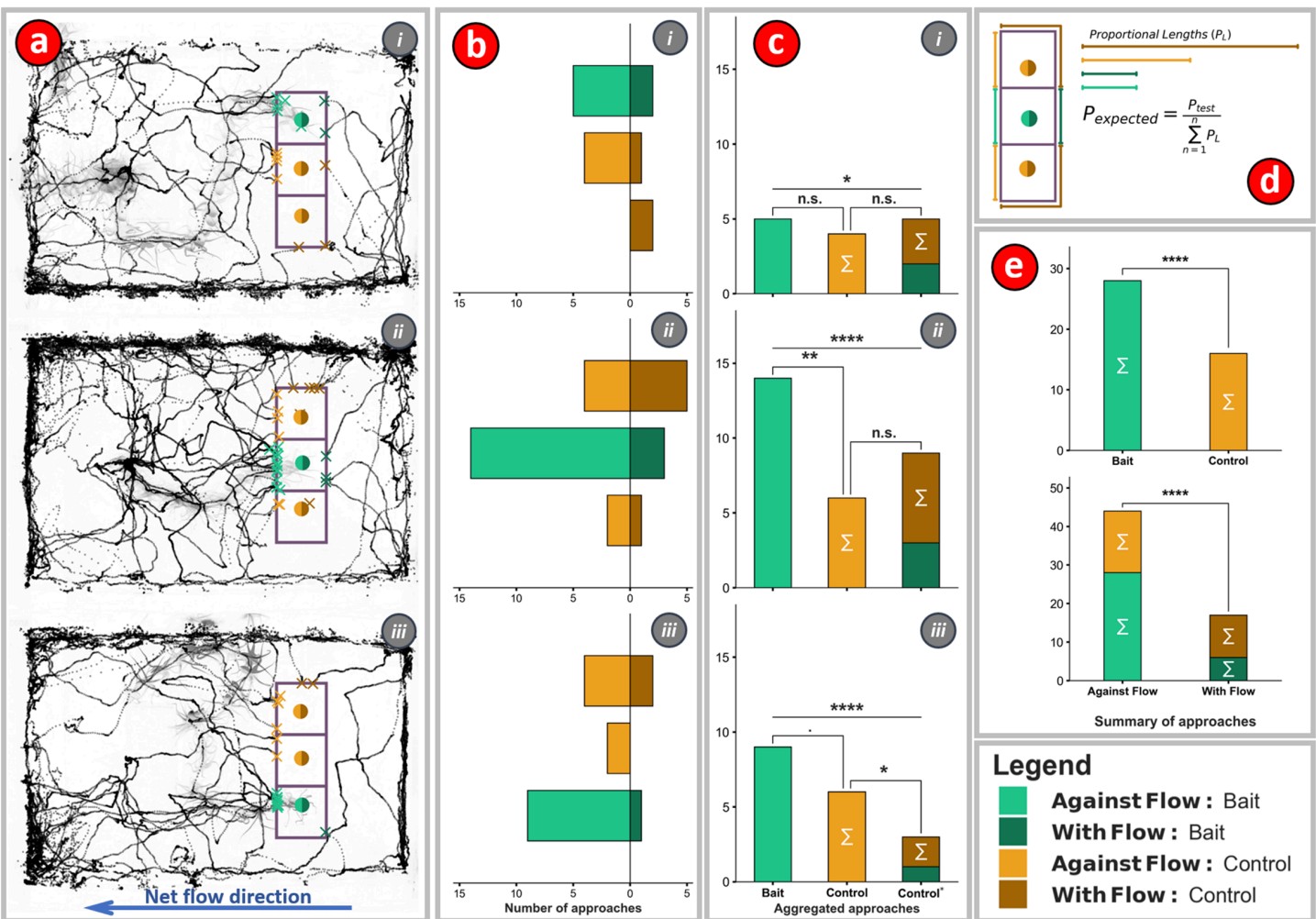

**Fig 4. (a) Octopus can discriminate between baited and unbaited stations in the dark.** Approach paths of the octopus for all approach sequences to the baited stations within the flume. Approaches are backcast ten minutes from the time of eating for each of the three different station configurations (i, ii, iii). The point at which an approach crosses over into a digitally superimposed 'target box' is marked by an x on the perimeter of the box. Baited stations are shown in teal/dark green while unbaited stations are shown in orange/brown. Lighter colors denote 'against flow' approaches in which the octopus swims upstream to the current and chemosensory plumes could be used for target detection. Darker colors show 'with flow' approaches when chemosensory tracking could probably not be used to locate the target. **(b)** Observed numbers of approaches sorted by their respective location, approach, and station configuration categories. **(c)** Histogram comparing aggregated approaches sorted by approach condition (with flow or against flow) and target (baited, unbaited/control). **(d)** Total lengths of target bounding boxes for all four approach and target conditions: 1. With flow - baited target, 2. With flow - unbaited target, 3. Against flow - baited target, and 4. Against flow - unbaited target (colors scheme shown in legend). In the null hypothesis, the number of approaches to a target is proportional to the total external perimeter of the bounding box containing the target (which excludes the common perimeter shared by two target bounding boxes). In other words, the approach probability density, or number of bounding box crossings per unit length is constant. **(e)** Histograms comparing aggregated number of all approaches to baited and unbaited stations (top), and against flow and with flow approaches (bottom). Statistical comparisons between different approaches were calculated using a binomial test, while comparisons to the null hypothesis (constant probability of approach over the entire target box) were calculated using a chi-squared test. ($\alpha = 0.05$, · $< 0.1$, * $< 0.05$, ** $< 0.01$, *** $< 0.001$, **** $< 0.0001$, n.s. : not significant).

the experimenter would begin placing three evenly spaced stations, one baited, and two unbaited, into the flume (Fig 4a) in one of three configurations ([bait, control, control], [control, bait, control], or [control, control, bait]). The octopus were given at least an hour to find the bait before the stations were changed. Bait consisted of a piece of fresh shrimp or crab dissected immediately before placing the stations in the flume. When placing the stations, the experimenter wore a very dim red head lamp with an extra red filter to see in the otherwise

dark tent. *O. rubescens*, like many cold water marine animals, have poor red-light vision. Visible markers on the flume floor were used to guide station placement. Location of all placed stations was measured with ImageJ (https://imagej.net/ij/) and the average location and placement errors were calculated. The placement error was low, having a range of about 2 cm around the marked position of the placement marker. The average locations were then used to define a digital rectangle which was used to set the end points of the feeding approaches. The octopus were wide enough that they could touch two stations simultaneously and would typically move between the stations after reaching one of them. The rectangle was split into three equally sized squares and assigned as either control or bait target areas based upon the station configuration (Fig 4a,d).

After either the octopus had eaten till satiation as determined by the experimenter, or a time between 00:00 – 02:00 had been reached, the octopus would then be returned to the holding tank and all stations removed from the flume. All trials were recorded continuously by video at 10 Hz and trials where the octopus ate were marked as successful. Sessions in which the octopus failed to reach the bait after an extended period were deemed unsuccessful as plume-tracking behavior was not observed. For the individuals that did consume the bait, the first station they approached was regarded as the primary target of their response. The successful sessions were then annotated with DeepLabCut, and the frame and place of feeding recorded. The location of the first crossing by the octopus into the target square was used to determine which station they had arrived at. Arrivals were then split into bait or control, and approach paths categorized as 'with flow' or 'against flow' depending on the side where the approach path crossed into the bounding rectangle (Fig 4a,d,e). An assumption was made that motion with the flow in the flume precludes the use of chemosensory plume-guided search. Motion trajectories of the octopus up to ten minutes prior to when the octopuses arrived at the stations and ate were then visualized (Fig 4a). Counts of approaches to the different sides of the rectangle were compared, first within the baited station configuration, and then between all stations aggregated. The total number of first passage events, when the octopus trajectory first crosses the boundary into the target rectangle, was divided by the perimeter of the target rectangle to obtain a probability per unit length. This linear probability density was used to set the expected ratio of observations for binomial and chi-squared tests.

## Single-station approach task

*Octopus rubescens* (5 male) were housed sequentially in the flume for a week and fed nightly from a baited station at a fixed upstream location in the flume. Only male octopus were found during collections for this experiment. A total of nine male *O. rubescens* were tested but four were excluded from analysis due to presumed age-related, senescence-suppressed appetites. Similar to the prior experiment, when placing the stations, the experimenter wore a dim red head lamp with an extra red filter to see in the dark tent. During their week in the experiment the octopus were monitored remotely and recorded continuously at 10 Hz under NIR light with a camera and fed exclusively at the fixed position baited station. Following the artificial sunset at 18:30 the experimenter would place the baited station into the flume, which gave the octopus at least an hour to find the food before the bait was changed to maintain its freshness. The bait was periodically changed until 00:00 - 02:00, when, depending on observed octopus activity, the experimenter would remove the baited station and finish the experiment for the night. Octopus in the flume showed nocturnal patterns of activity (App. 1, Fig 3). They also showed a strong preference for remaining close to the walls of the flume while active (App. 1, Fig 4), being near the wall 96.1% of the time while alert and active.

To determine the boundaries of the chemosensory plume used in the analysis of the single-station approach task, an artificial plume consisting of a mixture of 9-13 micron silica beads (SigmaAldrich 440345, Glass Spheres) and sea water was injected for three minutes into the flume via gravity at the location of the feeding station and imaged at 10 Hz using the same fixed camera used for the behavioral observations. The injected plume was visualized using a laser light sheet generated by a 50 mW green laser that was passed through a 70° Powell lens to form a laser light sheet with a uniform intensity profile. The light sheet was positioned at the height of the baited station, which was 2.5 cm above the flume floor. The three minute video was pixel-wise variance-projected and thresholded to produce a high-contrast image of the plume, and a polygon was manually drawn around the thresholded extent of the plume using Napari (Fig 5a) (https://napari.org/stable/). This polygon was then used for the analysis of all single-station experiments, and the intersection of the octopus motion trajectory with the perimeter of the polygon was taken to be the point at which any part of the octopus first makes contact with the plume. We assumed that the light scattered by the glass beads is a proxy for the concentration of the chemosensory chemicals. However, because we used 8-bit video recordings, the scattered intensity can only be subdivided into 255 light levels. On thresholding the light levels to set the edges of the plume, this dynamic range is further reduced. It is very likely that the octopus olfactory system is far more sensitive in that it has a much greater dynamic range for detecting chemical concentration than the camera has for detecting changes in light levels. Thus, it is likely that the simulated plume underestimates the possible extent of the actual chemosensory plumes encountered by the octopus.

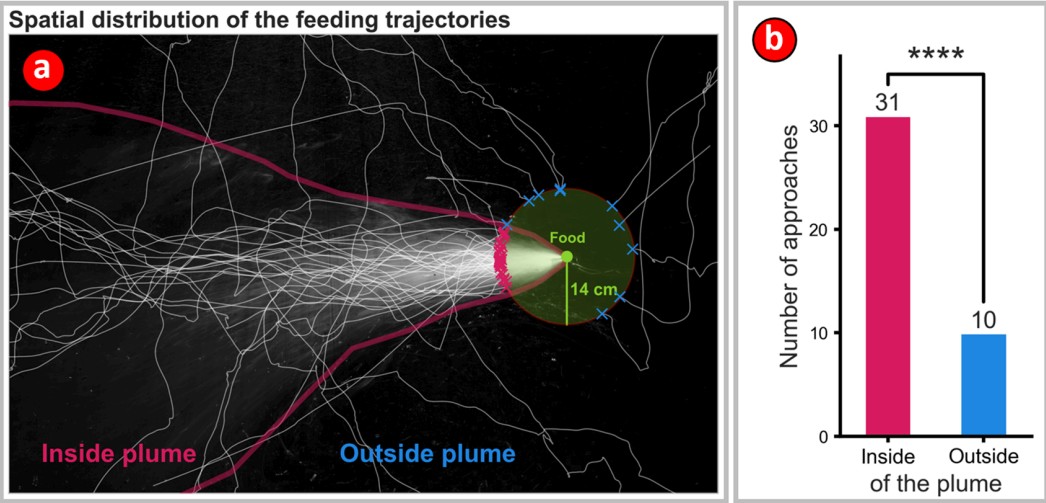

**Fig 5. Octopus preferentially approached the single baited station from within the plume. (a)** The thick magenta line shows the approximate extent of the plume. This boundary was determined visually using the time-averaged image of the simulated chemosensory plume (Fig 2d). A 'target circle' (green) centered on the baited station, and with a radius of 14 cm (the approximate maximum reaching distance for the octopus) was used to quantify the final approach angle of the octopus to the food target. The target circle and the plume boundary were then superimposed on the trajectories of all successful feeding events. The number of approaches to the target from the 'Inside plume' region are clearly seen to exceed approaches from the 'Outside plume' region. **(b)** Octopus preferentially approached the target along trajectories that clustered around the centerline of the chemosensory plume rather than from trajectories that lay outside the extent of the plume. (p = $2.6 \times 10^{-16}$, binomial test, expected ratio = 0.17).

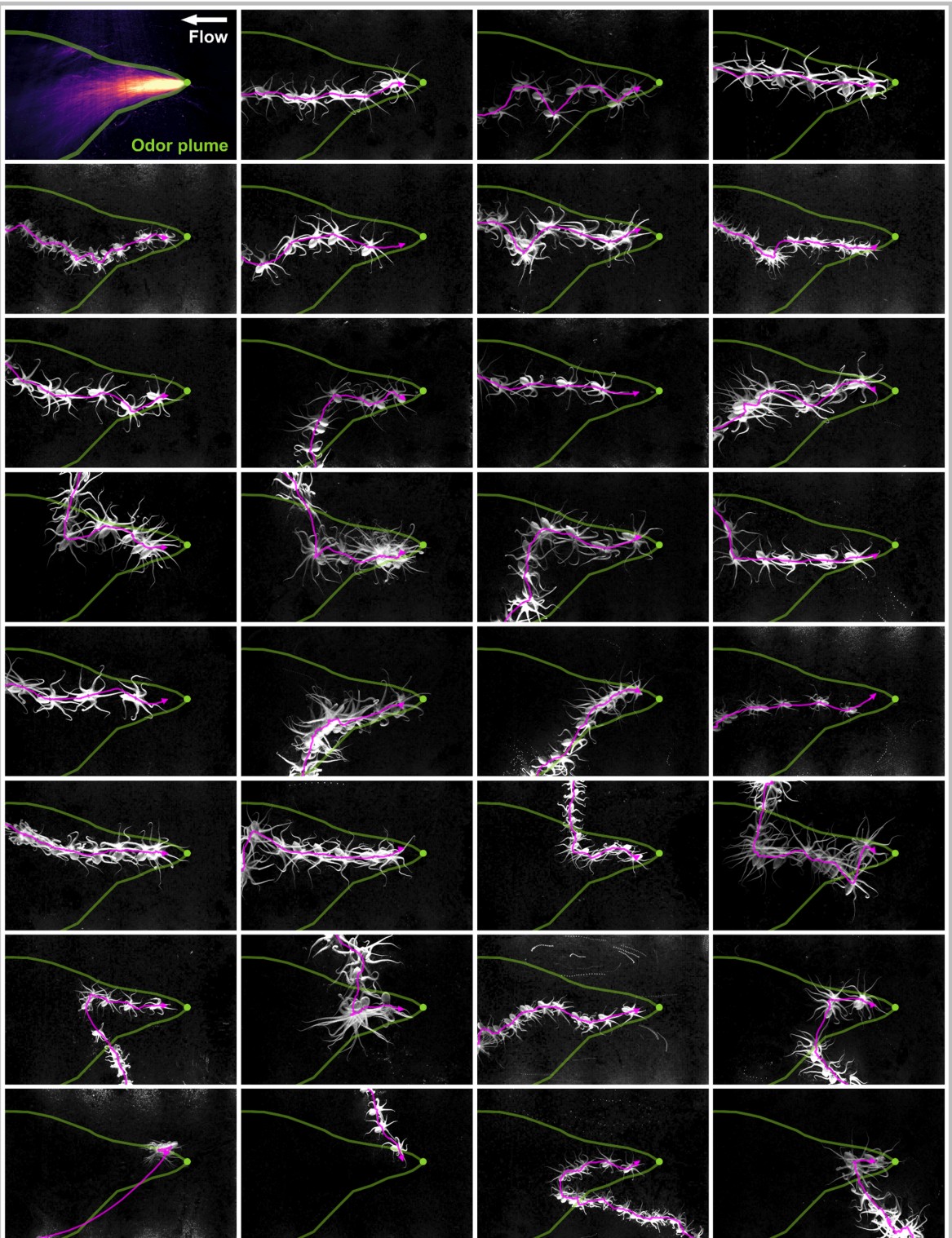

**Fig 6. Octopus displayed odor-gated rheotaxis behaviors during chemosensory plume-guided search.** Behaviors included cross-stream redirections, zig-zag casting motions, and surges. Approaches were made in many different body orientations relative to the bait, with minimal body rotation and a complex diversity of arm motions. The baited station location is shown as a green dot, and the approximate extent of the plume is shown in green. Water flow through the flume is from right to left. Octopus images are shown at three-second intervals, and the magenta line shows the trajectory of the body center, which is the midpoint of the line joining the two eyes (Fig 2).

### Segmenting chemosensory tracking feeding approaches

Video frames showing when the octopuses ate at the baited station were manually annotated and approaches from within the artificial plume were tracked automatically using the average eye position predictions (Fig 5). For the analysis of the feeding trajectories, a virtual target circle with a radius of 14 cm was defined around the station location. This distance is just further than what the octopus could reach easily. The octopus was considered to have reached the baited station when the midpoint between the eyes had crossed over into the circle, and all successful approaches ended in feeding. The polygon marking the extent of the plume was overlaid on the target circle, and the portion of the circle which overlaps with the plume was marked. As with the analysis of the three-target experiment, all events when the trajectory crossed into the target circle were counted and the total was divided by the circumference of the circle to obtain a linear probability density. This density was used in a binomial test to compare the ratio of trajectories where the crossing into the target circle occurred from within the plume, to those that were from outside the plume.

### Visualizing single-station differences in octopus motion between chemosensory tracking and controls

Octopus motions were classified into three categories: *Chemosensory tracking*, *Food not eaten*, and *No food*. The *Food not eaten* trajectories consisted of all the times when the baited station was in the flume and the octopus crossed into the plume, but did not eat the food, and that food was then not eventually eaten on another approach. The *No food* trajectories consisted of all the times when the octopus crossed into plume when no food was present in the flume and, consequently, there was no chemosensory plume to track (Fig 7c). The Food not eaten and No food trajectories were used as controls. (Fig 7c). For each of the three categories the body axis angles, heading offset angles, and nearest-arm offset angles were calculated for all frames (Fig 7) and were visualized in four ways. First, the motion sequences themselves were overlaid upon the plume with the body axis angle unit vectors shown (Fig 7a-c). Second, the evolution of the body axis angle was visualized as a polar plot with time as the radial distance from the origin and body axis angle as the polar angle (Fig 7d-f). Third, the distribution of the heading offset angle versus speed z-score was visualized (Fig 7g-i), and finally, the distribution of the nearest-arm heading offset angle as a function of the speed z-score was visualized (Fig 7j-l).

### Fast arm-aligned motions

Analysis of trajectories showed that the octopuses performed an interesting type of motion, which we have termed as fast arm-aligned motion (FAAM). During a bout of FAAM, the octopus stretches out an arm such that the base of the arm forms a straight line, and the octopus rapidly moves along this line in a lunging motion. It is as though the base of the arm forms a direction vector and the octopus rapidly translates its body in the direction specified by this vector. Visualization of the nearest-arm heading offset angle as a function of the normalized speed (speed z-score) showed that FAAMs were more frequent during chemosensory-tracking. Bouts of FAAM were segmented out of motion trajectories by filtering trajectories using the following criteria: (1) $1.5 \leq$ speed z-score $\leq 4.0$, (2) $-5° \leq$ nearest-arm heading offset angle $\leq 5°$, and (3) event duration $\geq 0.5$s. Fig 8b, c show the number of FAAM events and the percentage of time occupied by the FAAM events for the three classes of trajectories, namely *No food, Food not eaten*, and *Chemosensory tracking*. To further compare the differences between the three types of motion trajectories, one-second windows of

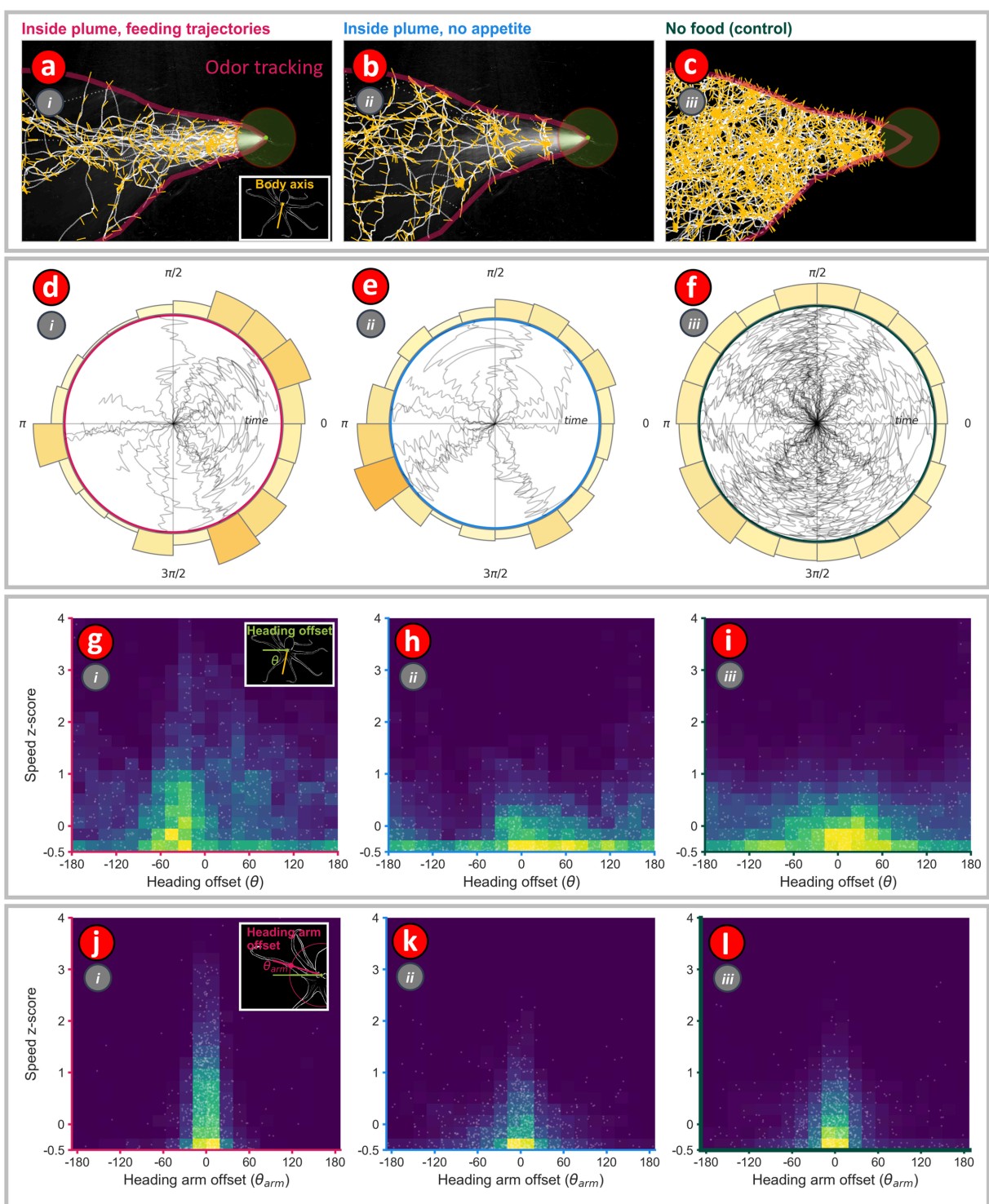

**Fig 7. The distribution of speed and nearest heading arm offset angles is different when octopus are chemosensory tracking. (a – c)** Octopus body center motion trajectories for three different conditions: (i) Inside plume approaches with feeding, (ii) Inside plume trajectories with no feeding, and (iii) No food. Overlaid on the trajectories are yellow 'hairs', drawn at three-second intervals, that show the body axis direction of the octopus during the motion. The location of the food is shown by a green dot, and the approximate extent of the chemosensory plume is shown in magenta. **(d – f)** Polar plots showing time evolution of the body axis direction of the octopus for the three different conditions. **(g – l)** In panes g - l, actual data points are shown as dots, while the density of dots is represented as a color-coded heatmap. **(g – i)** Relation between heading offset angle (angle between body-axis vector and velocity vector (shown in inset to pane g)) and speed z-score for all motion trajectories. **(j – l)** Relation between heading-arm offset angle (angle made by the leading arm with respect to body orientation vector (shown in inset to pane j) and speed z-score for all trajectories.

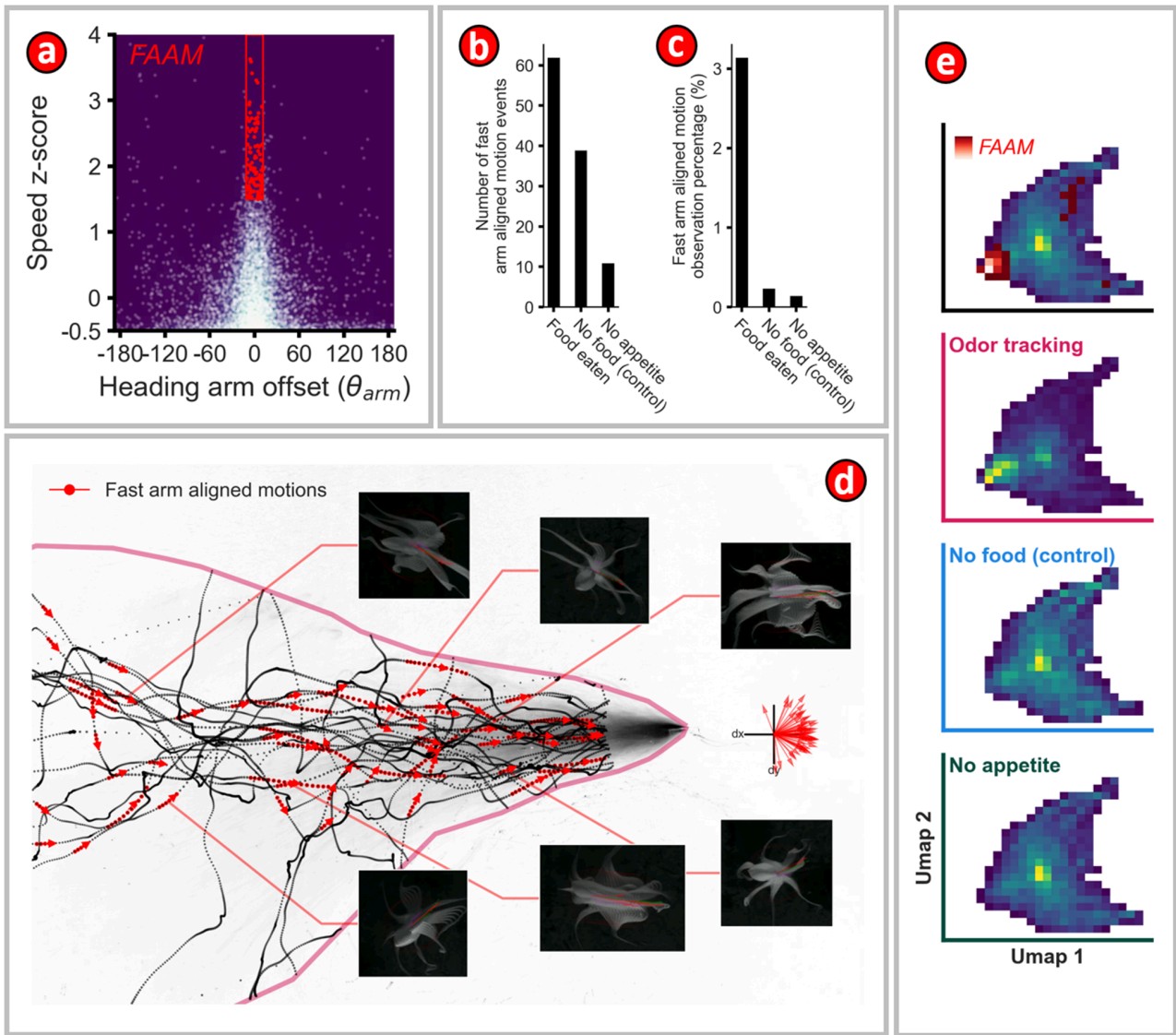

**Fig 8. Octopus did more fast arm aligned motions during chemosensory tracking.** (a) Relation between speed z-scores and heading-arm offset angles for all motion trajectories. The fast arm-aligned motions (FAAM) are shown in red. (b) Number of FAAM events for three different conditions. (c) Percentage of FAAM for different conditions. (d) Sequences of FAAM motions, shown in red, are superimposed on the 'in-plume' chemosensory-tracking motion trajectories. Arrowheads indicate the direction of motion of the octopus. Frames from six examples of FAAM motion sequences are shown in the insets. Octopus images are shown at 0.1 second intervals. The images show the complexity of the body shapes generated during a bout of FAAM. The red arrows point towards the arm most aligned with the velocity vector, which is shown in green. The direction of the body axis is shown with a faint magenta arrow. The quiver plot on the right shows the mean direction vector for all FAAM sequences for the in-plume feeding trajectories. The mean direction vectors are primarily clustered in the direction opposite to the mean flow direction. (e) All motion trajectories for the three different conditions were segmented into 1 second-long lengths and the corresponding speed z-score and heading-arm offset angles for these segments were combined and visualized using UMAP. The top pane shows the behavioral map for all conditions with the FAAM condition superimposed on it. The FAAM sequences are clearly seen to occupy a distinct region in the UMAP visualization.

the normalized nearest-arm heading offset angle and speed z-score values were jointly embedded into 2D UMAP space and the resulting 2D distributions were visualized as heatmaps. Fig 8e shows these heatmaps, which clearly show that FAAM bouts are much more frequent during chemosensory tracking.

## Results

### Octopuses use chemosensory tracking to locate food: three station discrimination task

In the absence of visible light the octopus approached the stations more often against flow than with flow to eat in a spatially non-random way (Fig 4e, p = 6.91×10⁻⁶, binomial test, expected ratio = 0.375). Then, when moving against flow the octopus went to the baited station over the unbaited one in a spatially non-random way (Fig 4e, p = 4.25 × 10⁻⁵, binomial test, expected ratio = 0.33). Given the size of the flume and the flow conditions within it these results support the hypothesis that octopus use chemosensory-plume guided search to locate food.

### Octopuses employ odor-gated rheotaxis and arm-aligned movement while navigating to food sources: Single station approach task

A total of 31 inside-plume feeding approaches and 10 outside-plume approaches were observed, which gave an average feeding event rate of 1.17 per night of the experiment. When approaching the single baited station in the absence of visible light the octopus approached more often from inside the plume than from outside it in a spatially non-random manner (Fig 5b, p = 2.5 × 10⁻¹⁶, binomial test, expected ratio = 0.17).

Odor-gated rheotaxis is a chemosensory-tracking strategy that is used by many animals and it primarily consists of two distinct types of motion: upstream surging, and casting. A 'taxis' refers to any motion that is directed in some way by the environment. Rheotaxis refers to motion that is oriented in a particular direction with respect to the flow of the fluid that the animal is immersed in. In the upstream surge phase, the animal rapidly moves upstream on encountering the chemosensory plume. When the animal loses contact with the plume, it begins casting, which is a zig-zag motion usually perpendicular to the mean flow direction. The purpose of casting is to regain contact with the plume. Thus, by alternating surging and casting bouts of motion, the animal can often navigate successfully to the chemosensory source. Surging and casting have also been compared to exploitation and exploration, respectively, as the upstream surge exploits the contact with the chemosensory plume, and a successful surge could result in the animal reaching the chemosensory source if it is directly downstream of it. When casting, the animal is now in an exploration mode as it is exploring its environment to detect the chemosensory signal again.

During approach the octopus displayed many characteristics of odor-gated rheotaxis such as cross-stream redirections towards the bait, surges, zig-zags, and pausing (Fig 6). The octopus approached the baited station in all body axis orientations but with a preference for motions with low heading offset angles (Fig 7a,d,g). This forward-facing motion preference was also observed during the control motions (Fig 7h,i). No obvious signs of bilateral reorientations towards the chemosensory source were observed, thus supporting the hypothesis that the octopus rely on their arms for chemosensory plume-guided search (Fig 7). During chemosensory tracking octopus displayed a roughly thirteen times higher rate of fast arm aligned motions (FAAM 3.1%) than the 'No appetite' (0.148%) or 'No Food' conditions (0.24%) with an average of two FAAM events detected during each of the 31 chemosensory-tracking sequences (Fig 8c). Additionally, the speed z-score and nearest-arm heading offset angle UMAP space occupancy was different for the octopus during chemosensory tracking than otherwise (Fig 8e). Thus, the octopuses appeared to use an chemosensory tracking strategy which involves reactive leading motions by the arms to stimuli. This supports the

hypothesis that the arms are an important chemosensory system for octopus chemosensory plume-guided search.

## Discussion

The unique neuroanatomical architecture of the octopus imposes a number of constraints on encoding sensory information within the brain. Despite a majority of the octopus's neurons existing within the arms and suckers, the afferent pathways from the arms to the brain communicate a small fraction of this information at any given time. Specifically, the arms' nervous system of an estimated 350 million neurons communicates with the brain through a bandwidth of an estimated 140 thousand afferent neurons [45–48]. This suggests that much of the sensory information gathered from the sucker rims, though dense with chemotactile receptors, is not centrally encoded. Evidence suggests that the orientation of mechanical irregularities encountered by suckers and proprioception representing the relative position of the arms and suckers are among the information not transmitted along this pathway [49–51].

Without these key details, the brain appears to rely heavily on local sensory integration to guide motor control. Efferent motor pathways originating from the brain broadly innervate large pools of motor neurons along the arms, and local sensory input appears to specify where amongst these regions the behavior will be activated [32]. These factors suggest that much of the circuitry underlying the arms' behavior is locally organized, a finding supported by several reports of behavior remaining intact even after an arm's connection with the brain is severed [47,51–57]. This localized organization of sensorimotor circuitry allows the arms to respond rapidly to sensory stimuli without input from the more distant brain. With the central role of the arms and suckers in foraging and prey capture, these fast, local responses are critical for the octopus's ecological success. Because chemical information is a strong indicator of prey, it would make sense that distance chemoreception is among the sensory systems locally integrated within the arms; specifically, the part of the octopus distance chemoreception system that is responsible for the localization of distant chemosensory sources.

Octopus have two separate external chemosensory systems, (1) the many suckers along their eight arms, and (2) their paired olfactory organs. Of these, the most obvious candidate for the chemosensory system responsible for dynamically guided behavior are the suckers, the rims of which are densely innervated with chemosensory anatomy. This hypothesis is supported by our finding that fast arm-aligned motions, and reactive lunging arm motions, happen around thirteen times more often during chemosensory-guided search than the 'no food' control condition (Fig 8b, c). The hypothesis is also supported by the lack of consistent bilateral reorientations to the plume. However, we do not yet know the complete pathways by which the complex stimuli of chemosensation are encoded and decoded within the suckers. We conjecture that the ring shape of a sucker may permit detection of the direction of both chemical and tactile stimuli through recurrent inhibition.

Octopus olfactory organs, which are the sensory epithelia that innervate the olfactory lobes, are located in miniature pits in the mantle crease. This is a poor location for active sensing, and their behavioral use by octopus is yet to be positively identified. To our knowledge, Messenger and Young [58] carried out the last attempt to directly modify the olfactory lobes and organs of octopuses to investigate their behavioral use. The study failed to find any evidence that the olfactory organs were chemosensory or that the lobes were involved in chemosensing. Squid olfactory organs are not pits and are more exposed anatomically for active sensing and have been shown to be used to initiate escape responses in response to chemical cues [59]. What seems most likely to us is that the octopus olfactory organs are important for modifying the physiology of motivation and reproduction but not guiding the

fast behaviors of search. It should be noted that both the olfactory lobes and the arms of the octopus have connections with the basal lobes, which are known higher motor centers in the brain of the octopus [58]. So, it is possible that both systems are involved in fast behaviors, but as noted in [58], it is not clear how one could test the olfactory organs in the absence of the arms.

It has been noted for many years that olfactory systems exhibit great similarity across phyla [3] in the architectural motif of glomeruli, a repeated pattern convergently evolved across many phyla. Chemosensory glomeruli are small, usually spherical, neural structures located within the olfactory bulb of the vertebrate brain. These structures are critical to smell. They serve as initial sites for processing olfactory information received from the environment. Each glomerulus receives input from olfactory sensory neurons expressing the same type of odorant receptor. In mammals these neurons are located in the nasal epithelium and detect airborne chemosensory signals. When a chemosensory molecule binds to its corresponding receptor on the surface of an olfactory sensory neuron, it initiates a signal that is transmitted to the olfactory bulb. Here, the axons of the sensory neurons converge onto a specific glomerulus, allowing the olfactory bulb to organize olfactory information based on the receptor type. This organization is a critical step in the brain's ability to identify and discriminate between thousands of different odors. Scientists have even gone so far as to hypothesize and computationally test the organization of glomeruli as an optimal anatomical motif for decoding chemosensory information [60].

However, the glomerular motif has yet to be positively described within any coleoid cephalopod [61] despite their being an ancestrally ancient and dominant marine species which co-evolved in intense competition with fish [62,63]. We posit that a glomerulus-like circuit may exist within the suckers of octopus and it has already been imaged and partially described by [64–66]. Namely, these are the encapsulated neuron circuits, which were hypothesized by [64] to be possible chemosensory amplification circuits, but missed the connection to other olfactory systems. In the mammalian nervous system, encoding of sensory information is primarily done within the olfactory bulb. In the octopus, chemosensory information encoding would be done by circuits in their arm nerve cords and sucker rims. However, instead of two central olfactory bulbs, the octopus has many, possibly even one for each sucker. These are the neuropil bulges along the arm nerve cord neuropil associated with each sucker [32]. These bulges are responsible for encoding more than just distant chemosensory cues, but also tactile and taste cues. This information encoding occurs rapidly enough for both danger and grabbing reflexes to exist, but also for long-range chemosensory tracking reflexes. Given the enormous amount of information compression that occurs between the sensory input to a sucker and signal decoding in the central nervous system, when combined with the need for optimization of neural growth in the arms, the chemosensory circuits responsible for discriminating distant chemical cues in the octopus arms are likely to be fundamentally different from other organisms and worthy of expanded scientific effort to understand them.

Octopus are considered by many to be the most intelligent invertebrate with capabilities similar to that of evolved vertebrates such as rodents, cats, and dogs. Octopus are known to exhibit complex spatial navigation behaviors in the wild [13], although limited success eliciting these behaviors has been reported in the laboratory [67]. Most terrestrial predators which exhibit similar behaviors are known to be able to incorporate chemosensory information into their world model [2]. Many species of octopus are known to forage over large distances in the wild in poorly lit benthic environments and then return to specific locations [68,69]. It would be surprising if parallel mapping of chemosense onto their world models was not a capability these octopus possessed.

It is useful to contrast octopus long-range chemosensation with how rodents use airborne chemosensory cues to navigate environments, avoid predation, and acquire resources such as food, often under conditions with limited visibility [70]. Deer mice, for example, detect food using olfactory rather than visual cues [70]. While the abilities of rodents have been appreciated for decades, more recent experimental and computational work addresses the mechanistic basis for these behaviors [4,71–73]. Findings from recent mechanistic work regarding chemosensory navigation by rodents using airborne chemosensory cues highlight the role played by active sensing (sniffing) and demonstrate that rodents adaptively sample the air at informative locations during a simple left/right decision task based on airborne chemosensory cues [73]. Recent work that expands into more complex environments supporting turbulent flow and the development of chemosensory plumes suggests that rodents efficiently use plumes to navigate, being significantly more capable than both simulated or robotic systems executing simple chemosensory tracking algorithms [74]. Perhaps long-term memory, which rodents balance with the sensory cues provided by plumes, supports this increase in tracking efficiency [75,76]. The reliance on long-term memory distinguishes rodent plume-based tracking strategies from strategies found in flying insects, which act primarily as detectors and trackers, and, although they show some short-term history dependence, have a strong reactive component [77].

Octopus plume-tracking shows both similarities and instructive differences from that of rodents. Both octopus and rodents use active sensing to interrogate a chemosensory plume. Rodents perform saccade-like sweeping head movements along with repetitive, high frequency sniffing that, when combined, appear to enable them to sample a greater volume of air while searching for a packet of odor dispersed within a plume. Similarly, we have found that octopuses engage in what appears to be an active sensing behavior, which we have called fast arm-aligned motion (FAAM). Our belief is that these are reactive motions by the arms in response to contacts with the chemosensory plume. Rodents memorize complex distributions of food rewards and show a preference for using this memory over following airborne chemosensory cues [76]. It is reasonable to ask whether octopuses are similarly memory-reliant. However, the design of our experimental tests would not have been able to observe a similar ability in octopuses. As a cold water species, *O. rubescens* only eats once or twice a night, if at all, even given the small size of food rewards required by the baited station. In contrast, a rodent with a similarly simple and repetitive food location could take many more small food rewards in the same time period; and with many more trials, would have quickly begun to easily locate the food without waiting for plume guidance, exhibiting ballistic approaches to the learned, single possible location of the food, with only a few days of training. To know whether or not octopus have similar memory based capabilities as rodents, new tasks, and new ways of training them for these tasks need to be invented.

## Conclusion

We assert that our understanding of how the octopus integrates chemosensory information for long-range chemosensory navigation has to start at the level of an individual sucker. Once that is understood, how groups of suckers integrate information together can then be understood, followed by how this information is pooled at the level of the entire arm and sent to the brain. It is perhaps only then that we will be able to unravel how sensing becomes motion. Given the proximity of taste, touch, and smell in the octopus, it would not be surprising if the different sensory modalities are encoded together locally within the arm. We suggest that future studies should explicitly consider this possibility.

We posit that studies of octopus chemosensory search using chemosensory cues are a good behavioral paradigm to investigate how information between the suckers, arms, and the brain of octopus are integrated at the organismal level. This is because such experiments can involve the full sensory range of the octopus and the behavior can be imaged as well. Chemosensory search by octopus has received a lot of attention [5–14], but this large body of work has almost solely focused on the tactile searching phase, which is the final stage of olfactory visual search. This has left largely unaddressed the question of how chemosensation is used when chemicals are sensed while they are dispersed in a flowing fluid medium and not when they are present on a solid surface; which is the focus of our work.

Additionally, we anticipate that it will be found that some species of octopuses may have many chemosensory capabilities similar to that of vertebrates, including mapping of chemosensory information into their world model and rapid place-learning.

Ours is the first study to successfully use detailed analyses of arm motions and body orientation to show that during plume tracking only, octopuses often followed a lead arm, and were thus following the sensory organs located in the suckers. There is a lot left unknown about octopus chemosensory plume-guided search. Unsurprisingly, significant complexity of arm motion by the octopus was observed during all motions, controls, and while chemosensory tracking, limiting our analysis to a specific subset of behavior.

We believe that the main technical problem to be solved is to develop the ability to identify and track each individual arm of an octopus while simultaneously imaging the chemosensory plume the octopus is interacting with naturalistically to find its source. We believe that this problem can be solved in certain trainable species. However, until it is, the study of octopus chemosensory-plume guided behaviors will be challenging. Such a solution might involve imaging filaments of chemosense-activating chemicals with an infused infrared contrast agent contacting the arms of the octopus using near infrared laser imaging, all while the octopus is engaged in a training paradigm similar to a rodent chemosensory water or food restriction task [71], with small titrated automated rewards delivered to encourage them to interact with the artificial plume. This would be similar to how rodents can be trained to interact with ethanol plumes [71]. No group has successfully demonstrated such a system of fully automated small titrated rewards for octopus on any task. We expect such a paradigm could be expanded to help understand many other facets of how octopus use chemosensation as well as other sensory modalities. The algorithms driving chemosense-following behavior are likely to involve a combination of local reactive arm-control reflexes and memory, and might lead to the discovery of a different, distributed, bottom-up-driven way of doing chemosensory plume-guided search than in canonically studied model systems which involve top-down neural control.

## Supporting information

**S1 Fig. Estimating octopus pose with DeepLabCut highlights the challenge of tracking the arms. (a)** Performance of the full frame model on tracking the eyes of the octopus. Eye tracking mean squared error test performance plateaued at 6 mm, a size smaller than the eyes of the octopus and on par with human error. **(b) Performance of the cropped-frame model on tracking the eyes and arms of octopus** Arm tip tracking mean squared error plateaued at 18 mm, a size far larger than the arm tips. Cropped frames were passed to DeepLabCut for arm tip annotations for two reasons, (1) the arm tip feature size is very small and the full frame resolution is large (1920x1200), and (2) humans find it much easier to annotate arm tips in cropped frames. **(c)** Flow chart of image processing sequence. **(d) Visualizing body part velocities highlights the poor quality of arm tip tracking** Velocity traces for each of the

arm tips were thresholded to remove outliers and then smoothed using a Savitsky-Golay filter. Despite this the data showed large (both positive and negative), unphysical values for the arm tip velocity (red arrows) due to tracking errors. **(e)** DeepLabCut traces for all body parts for an example chemosensory tracking sequence. Solid thick lines are the smoothed traces while the thin faint lines of the same color are the raw traces.
(TIF)

**S2 Fig. Normalizing octopus speed distributions with z-score. (a)** Kernel density estimation (KDE) of the speeds of the five different octopuses used in the single station approach task. The kernel density estimates were calculated after smoothing the velocity traces with a Savitzky-Golay filter. **(b) Kernel density estimation of octopus speed z-scores** Both (a) and (b) used a KDE bandwidth of 0.05.
(TIF)

**S3 Fig. *Octopus rubescens* in captivity showed nocturnal patterns of activity.** Location of 10 different alert and active octopus, held in the flume at different times, continuously recorded at 10 Hz for a total of 66.9 days of observations. A pair of experimenters exhaustively annotated when the octopus were visibly alert and active in all days of recording. The large size of the flume and relatively low-resolution of the footage made many subtle behaviors unobservable. Consequently, octopus that were stationary for long periods, regardless of quiescent state, were annotated as not alert and not active. **(a) Raw unfiltered DeepLabCut mean eye locations of the octopus split by hour of the day** Plot boundary colors indicate night, dawn, day, and dusk. **(b) Fraction of the time each hour that the octopuses were alert and active** The data points are the averages for each octopus obtained after averaging over the entire video dataset. The thin curves show the variation of the averages for each individual octopus across the day. The thick line is the data averaged over all octopuses.
(TIF)

**S4 Fig. Alert and active octopus in the flume had a strong preference for remaining on the wall. (a)** Raw unfiltered DeepLabCut predictions of mean eye location for ten different alert and active octopus in the flume (S3 Fig.). Two-dimensional kernel density estimation (KDE) was used to show the distribution of space occupancy. A blue rectangle within each KDE plot delimits the on and off the wall boundary. Amount of in and out time measured for each individual is shown in red text. **(b)** Averages of time spent in the open and on the wall by each octopus. On average the octopus spent 96.1% of their time on the wall while alert and active.
(TIF)

**S5 Fig. Trace of custom image processing algorithm with and without Savitzky-Golay filtering of eye coordinate predictions. (a)** Trace before Savitzky-Golay (SavGol) filtering of the DeepLabCut eye coordinate predictions. Due to the large radius of the bounding circle used to define arm positions (see Fig. 3d), small oscillations due to prediction error in the eye oordinates result in large, spurious fluctuations of the arm positions. **(b)** Trace after Savgol filtering. High frequency oscillations are noticeably reduced.
(TIF)

**S6 Fig. Schematics of tasks (a)** One-station approach task schematic. Dashed magenta rectangle is the bounds of segmented trajectories for the task. Solid magenta line shows the length from the downstream edge of the rectangle to the food station. The red circle segmented the end trajectories that ended with feeding, the circle's radius is roughly the length of the octopus arms. **(b)** Three-station discrimination task. Dashed red squares are the segmentation edges of the three station task stations. Small green squares are the location of the stations.

Long magenta line is the distance from the downstream rock tray to the stations. Small magenta lines is the distance between the stations. Inner green rectangle and triangle are the rock tray and den respectively. In both **(a)** and **(b)** the outer green rectangle visualizes the dimensions of the arena.
(TIF)

**S7 Fig. Elaspsed time for octopus single station trajectories.** In-plume periods are indicated by orange highlights. **(a)** All feeding approach trajectories for the single station experiment. Black hatches indicate the moment when the eyes of the octopus crossed into the circle before feeding. **(b)** All 'no appetite' trajectories where food was present but the octopus did not eat. **(c)** A representative sample of control trajectories in which no food was present. All panels **(a-c)** were designed with the same vertical and horizontal extents so that direct comparisons could be made by eye.
(TIF)

## Acknowledgments

We thank all the staff and researchers at the Friday Harbor Laboratories who made this project possible. We especially acknowledge Joseph Ullmann, John Dorsett, Theresa Phillips, and Julia Kobelt for invaluable assistance. We also thank Michael Griffin for assistance with flow visualization experiments.

## Author contributions

**Conceptualization:** Willem Lee Weertman, Venkatesh Gopal, Dominic M. Sivitilli, David Scheel, David H. Gire.

**Data curation:** Willem Lee Weertman.

**Formal analysis:** Willem Lee Weertman, Venkatesh Gopal.

**Funding acquisition:** Willem Lee Weertman.

**Investigation:** Willem Lee Weertman, Venkatesh Gopal.

**Methodology:** Willem Lee Weertman, Venkatesh Gopal, David Scheel.

**Project administration:** Willem Lee Weertman, David Scheel, David H. Gire.

**Resources:** Willem Lee Weertman, David Scheel, David H. Gire.

**Software:** Willem Lee Weertman, Venkatesh Gopal.

**Supervision:** Venkatesh Gopal, David Scheel, David H. Gire.

**Validation:** Willem Lee Weertman.

**Visualization:** Willem Lee Weertman.

**Writing – original draft:** Willem Lee Weertman, Venkatesh Gopal, David Scheel, David H. Gire.

**Writing – review & editing:** Willem Lee Weertman, Venkatesh Gopal, Dominic M. Sivitilli, David Scheel, David H. Gire.

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
