## [Decision Letter · Decision Letter 0]

28 Apr 2025

PONE-D-25-06702Octopus can use odor plumes to find foodPLOS ONE

Dear Dr. Weertman,

Thank you for submitting your manuscript to PLOS ONE. After careful consideration, we feel that it has merit but does not fully meet PLOS ONE’s publication criteria as it currently stands. Therefore, we invite you to submit a revised version of the manuscript that addresses the points raised during the review process.

We look forward to receiving your revised manuscript.

Kind regards,

Alon Gorodetsky

Academic Editor

PLOS ONE

“WLW was significantly aided by the support of the Beatrice Crosby Booth Endowed Scholarship at Friday Harbor Laboratories, Alan J. Kohn Endowed Fellowship, and crowdfunding on Experiment.com.

VG would like to gratefully acknowledge generous support from the Hyslop-Shannon Foundation.

DSc acknowledges support from an Indigenous One Health development grant from an Alaska Native & Native Hawaiian Serving Institutions Program of the U.S. Department Of Education.”

5. Please update your submission to use the PLOS LaTeX template. The template and more information on our requirements for LaTeX submissions can be found at http://journals.plos.org/plosone/s/latex.

Additional Editor Comments:

Please thoroughly address the concerns of all thee reviewers, and specifically, note and address the concerns about terminology from Reviewer 1 with regard to "odor" versus "chemical cue".

Reviewers' comments:

Reviewer's Responses to Questions

**Comments to the Author**

1. Is the manuscript technically sound, and do the data support the conclusions?

Reviewer #1: Partly

Reviewer #2: Yes

Reviewer #3: Yes

2. Has the statistical analysis been performed appropriately and rigorously? 

Reviewer #1: I Don't Know

Reviewer #2: Yes

Reviewer #3: Yes

3. Have the authors made all data underlying the findings in their manuscript fully available?

Reviewer #1: Yes

Reviewer #2: Yes

Reviewer #3: Yes

4. Is the manuscript presented in an intelligible fashion and written in standard English?

Reviewer #1: Yes

Reviewer #2: Yes

Reviewer #3: Yes

5. Review Comments to the Author

Reviewer #1: The paper titled "Octopus can use odor plumes to find food" is an interesting paper showing that octopuses can perform chemical-plume-guided navigation, a behavior previously known in many animals but never before observed in octopuses. The study shows that octopuses can use "odor" plumes to track and find food, demonstrating that this behavior is not limited to animals with more typical olfactory organs but also occurs in cephalopods like octopuses. While the methods used as well as the collected results are nice and intriguing, the authors should take into account some important works that they ignored.

To avoid further bias in the literature regarding chemoreception, the authors must change the word "odor" to chemical or chemical clue throughout their paper. To understand why, they should carefully read and quote:

Taste and Smell: a unifying chemosensory theory. by Mollo et al., 2022 https://doi.org/10.1086/720097;

Sensing marine biomolecules: smell, taste, and the evolutionary transition from aquatic to terrestrial life. by Mollo et al., 2014 doi:10.3389/fchem.2014.00092;

Taste and smell in aquatic and terrestrial environments. by Mollo et al., 2017 DOI: 10.1039/c7np00008a;

Cephalopod Olfaction. by Di Cosmo and Polese 2017 DOI: 10.1093/acrefore/9780190264086.013.185.

Furthermore, for the same reason on why they must use chemicals instead of odor, it is essential that the "odor" or better the chemical clues (molecules) used in the experiment must be clearly specified in the material and methods section.

The discussion must also be reformulated in an evolutionary key accordingly.

I believe that the suggested changes will give a pivotal new and more interesting vision of the results obtained that will go further the already incredible technical work done.

As it is now the paper will contribute to confusion in the chemical perception literature and so it must be revised before publication

Reviewer #2: This study by Gire and colleagues is a wonderful and elegant demonstration of olfactory behavior in the octopus. They provide rigorous evidence that octopuses do indeed follow odor plumes, and quantify the behavioral motifs that the animals use to achieve this, including active sensing. The study thereby provides a strong experimental foundation for a phenomenon that had largely been described anecdotally. In addition to the new insight it provides into olfactory processing across the animal kingdom, this will provide a basis for the study of the underlying neural mechanisms in octopus.

I have only a few minor points for presentation.

1. There is no information on the time required to reach the target, which would be useful to get a sense of how efficient the animals are.

2. I am not sure the figures based on previous anecdotal evidence (Fig 1A-C) are useful here, except to provide contrast with the rigor of the current study, as they aren’t really data per se.

3. It would be helpful to have a schematic for the two behavioral setups, including length scale.

4. I wasn’t sure what the “sigma” symbol on Fig 4 signifies.

5. Fonts in many figures are almost too small to see.

Reviewer #3: This is a nicely written report of a very interesting and well designed study. The analysis is impressive and should be of use to other groups working on non-traditional models. Overall I think the study is novel, valuable and interesting and certainly worth publishing. Whilst I have a number of suggestions, they are all minor.

Introduction: "Efficient use of chemical cues for search and way-finding is an evolutionary pressure experienced by most organisms". Most organisms are not animals, so this should be rephrased

Figure 1 d: a clearer description of what the diagram here shows would be helpful; first, that this represents a model or a hypothesis, not actual data (at least I don’t think it shows actual data, either way this should be clarified), and second, that the fine black lines represent a theoretical odor plume and the red line represents the animals’ proposed movement under the three different strategies.

Introduction: "Some mollusks use chemoreception in predator avoidance, although this has yet to be demonstrated for octopuses. For example, both scallops (Speiser and Wilkens, 2016) and gastropods (See Edgell (2010) and references therein) increase escape responses in the presence of chemicals associated with nearby predator activity." The authors may wish to add Howard et al., 2019 here, which demonstrates suppression of feeding activity in the presence of olfactory predator cues in squid.

In the introduction the authors state that chemosensing in odor plumes has been studied in nautiluses only, which is not entirely correct. While they cite Chase and Wells, the clearest example of probable odor-gated rheotaxis in octopus, it is erroneously characterized as a study of chemotactile sensing. Notably, the results in this older paper report “arm waving” behavior, precluding the sensing only of substrate-bound molecules, and what would seem to be the same behavior as the FAAM movement describe here. “Other arm movements were more directed; these were extensions of the arms, which often preceded a locomotion oriented primarily upstream.”

Figure 7 is hard to understand. It looks like there are many more data points represented in the 7c (no food) panel, which makes any inference about differing trajectories very hard to interpret. I assume this is related to the plotting of positions every 2 seconds (although this is stated in the caption for Figure 6 and not figure 7, I assume it is the same), which would then imply that the octopuses moved more slowly in the absence of a directional odor cue. This makes sense but if the point of the figure is to show the more aligned trajectories (i.e., lower heading angles) in the presence of food, this really cannot be discerned from panels a-c.

Figure 8d. There are three different colors shown in the individual images; red arrows but also green and magenta arrows. An explanation for these colors should be given in the figure caption.

Figure 6 shows an interesting observation that the authors don’t really address; from these trajectories it does not appear that there is a significant change in movement speed as the octopuses get closer to the target. Change in velocity is a hallmark of close-distance olfactory search in many species, and it would be interesting for the authors to consider this variable in their analysis and discussion. Although it is hard to make clear determinations from figure 6 there certainly does not appear to be a consistent switch in strategy close to the target. Given the emphasis throughout the manuscript on the difference between octopuses and other animals where olfactory search is more deeply studied, this point could be elaborated.

Discussion and conclusions: the authors somewhat ignore one of the most interesting aspects of octopus neurobiology until the opening section of their conclusion, which is the possibility of central processing of odorant information that is received by two different arms. While their point that understanding single-sucker processing is a necessary precursor to building up a model of whole-arm or whole-body processing is fair, both the experiment design and the analysis pipeline really focus on whole-body aspects of tracking behavior, with little emphasis in the analysis of looking at single-sucker behavior. It is known that the olfactory lobes receive inputs and send outputs to the basal lobes where pathways from the arms are found, so it is worth discussing this aspect of octopus neurobiology. The behaviors they describe also raise the intriguing possibility of a somatotopic map of the arms informing the directional search behavior; the apparent absence of central brain somatotopy in cephalopods is of great interest to comparative neurobiologists, as is the apparent absence of glomeruli-like structures in the olfactory lobe itself. A brief discussion of central brain pathways and implications for central processing would broaden the general appeal of the paper. A large section of the discussion is devoted to an explicit comparison between rodents and octopuses, which is perhaps not the most informative comparison.

6. PLOS authors have the option to publish the peer review history of their article (what does this mean?). If published, this will include your full peer review and any attached files.

Reviewer #1: **Yes: **Gianluca Polese

Reviewer #2: No

Reviewer #3: No

---

## [Author Response · Author response to Decision Letter 1]

30 Jun 2025

Please see 'Response to Reviewers.pdf' document

---

## [Decision Letter · Decision Letter 1]

30 Jul 2025

Octopus track chemosensory plumes to find food

PONE-D-25-06702R1

Dear Dr. Weertman,

We’re pleased to inform you that your manuscript has been judged scientifically suitable for publication and will be formally accepted for publication once it meets all outstanding technical requirements.

Kind regards,

Alon Gorodetsky

Academic Editor

PLOS ONE

Additional Editor Comments (optional):

Reviewers' comments:

Reviewer's Responses to Questions

**Comments to the Author**

1. If the authors have adequately addressed your comments raised in a previous round of review and you feel that this manuscript is now acceptable for publication, you may indicate that here to bypass the “Comments to the Author” section, enter your conflict of interest statement in the “Confidential to Editor” section, and submit your "Accept" recommendation.

Reviewer #1: All comments have been addressed

Reviewer #2: All comments have been addressed

2. Is the manuscript technically sound, and do the data support the conclusions?

Reviewer #1: Yes

Reviewer #2: Yes

3. Has the statistical analysis been performed appropriately and rigorously? 

Reviewer #1: Yes

Reviewer #2: Yes

4. Have the authors made all data underlying the findings in their manuscript fully available?

Reviewer #1: Yes

Reviewer #2: Yes

5. Is the manuscript presented in an intelligible fashion and written in standard English?

Reviewer #1: Yes

Reviewer #2: (No Response)

6. Review Comments to the Author

Reviewer #1: The revised manuscript marks a milestone in the study of chemoreception in octopods. These cephalopods live close to the seabed and are therefore guided by chemical stimuli both dissolved in water and attached to the substrate. Their work sheds light on how these animals follow chemical cues dissolved in the aqueous medium.

Reviewer #2: (No Response)

7. PLOS authors have the option to publish the peer review history of their article (what does this mean?). If published, this will include your full peer review and any attached files.

Reviewer #1: No

Reviewer #2: No

---

## [Editor Report · Acceptance letter]

PONE-D-25-06702R1

PLOS ONE

Dear Dr. Weertman,

I'm pleased to inform you that your manuscript has been deemed suitable for publication in PLOS ONE. Congratulations! Your manuscript is now being handed over to our production team.

Kind regards,

on behalf of

Dr. Alon Gorodetsky

Academic Editor

PLOS ONE